# Effect of the Strength of Attraction Between Nanoparticles on Wormlike Micelle-Nanoparticle System

**Mubeena Shaikh**

IISER-Pune, Dr. Homi Bhabha Road, Pune 411008, India; mubeena@students.iiserpune.ac.in;
Tel.: +91-913-0731-891

**Abstract:** The nanoparticle-Equilibrium polymer (or Wormlike micellar) system shows morphological changes from percolating network-like structures to non-percolating clusters with a change in the minimum approaching distance (EVP-excluded volume parameter) between nanoparticles and the matrix of equilibrium polymers. The shape anisotropy of nanoparticle clusters can be controlled by changing the polymer density. In this paper, the synergistic self-assembly of nanoparticles inside equilibrium polymeric matrix (or Wormlike micellar matrix) is investigated with respect to the change in the strength of attractive interaction between nanoparticles. A shift in the point of morphological transformation of the system to lower values of EVP as a result of a decrease in the strength of the attractive nanoparticle interaction is reported. We show that the absence of the attractive interaction between nanoparticles leads to the low packing of nanoparticle structures, but does not change the morphological behavior of the system. We also report the formation of the system spanning sheet-like arrangement of nanoparticles which are arranged in alternate layers of matrix polymers and nanoparticles.

**Keywords:** self-assembly; polymer nanocomposites; polymer templating; equilibrium polymers; Wormlike micelles; mesoporous structures; bottom-up approach

## 1. Introduction

Nanostructures have a wide range of applications in energy devices [1,2], optoelectronic devices [3], drug delivery [4–6], cosmetics [7,8], food [9,10] and novel functional materials [11,12]. To produce nanostructures, the bottom-up approach has recently become a cost-effective and easy method in the nanofabrication industry [13,14]. Using polymeric matrices to assemble nanoparticles is one of the prominent methods in bottom-up approach [15–18], e.g., production of various nanostructures using di-block copolymers matrix [19]. However, tailoring nanostructures with a precise control over their size and shape is a challenge in nanofabrication industry.

Polymer nanocomposites (PNCs) form an active research field with a variety of theoretical and simulation techniques to get an insight into the structure-property relationship. The early equilibrium theoretical studies [20,21] integrate out the polymer degree of freedom while recent theories have be developed to take into account the degrees of freedom of both the particles and the polymers. Fredrickson and co-workers [22,23] used a self-consistent field theory framework to study the nanoparticle assembly in a self-assembled block copolymers [24–26] and a more generalized version was presented later by Riggleman and co-workers [27,28]. This was further developed by Balazs [29–31] and Frischknecht [32–34] to apply it to both homo and block copolymers matrices. There also exists simulation techniques developed by de Pablo [35–37] or a soft particle approach by Bolhuis and co-workers [38–40]. Despite such advances in the filed of PNCs, there exists significant gaps in understanding the PNC systems and their structure–property relationship.

In this paper, we employ equilibrium polymeric matrix (or Wormlike micellar matrix) to self-assemble nanoparticles into various kind of structures and investigate the effect of the strength of nanoparticle interaction. Nanoparticles with their high surface to volume ratio make it difficult to disperse them in a polymeric matrix. Therefore, nanoparticles are often grafted by polymers or have a surface modification to get a homogenous dispersion [41]. The assembly of such grafted nanoparticles in a polymeric matrix depends on the ratio of the size of nanoparticle core to the grafted polymer [42], polydispersity of the graft lengths [43,44], distribution of grafted chains on the nanoparticle surface [45], the grafting method [46], etc. Apart from the dispersion of nanoparticles in a matrix, it is also important to study the effect of interaction between nanoparticles to get a precise control over tailoring nanostructures with desired properties and shape.

The nanoparticles self-assemble in an equilibrium polymeric matrix to give rise to various kind of structures, viz. mesoporous networks, nanorods and nanosheets [47]. With an increase in minimum approaching distance between nanoparticles and polymers (EVP, excluded volume parameter), a morphological transition of nanoparticles from network-like structures to individual clusters has been shown in a previous study [47]. The study also indicates that we can control the anisotropy of the nanoparticle clusters by tuning the density of the matrix. In this paper, we report the shift in the values of the EVP required for the structural change of nanoparticles, as a result of the change in the strength of nanoparticle interaction. We also report a decrease in the packing of nanoclusters with a decrease in the strength of attractive interaction between them. Moreover, the formation of the system spanning nanoparticle sheets is also observed.

## 2. Model and Method

### 2.1. Modeling Wormlike Micelles

The model used in this paper is the same as the model used in [47,48] which is a modified version of the model presented in [49]. According to this model, the Wormlike micellar chains are coarse-grained as a chain of spherical beads. Each spherical bead (here called as monomer) in the model is assumed to represent a group of amphiphilic molecules at the mesoscopic scale. All the chemical details are ignored here and only the relevant details to describe behavior at the mesoscopic scale are considered. The schematic diagram of the model is shown in Figure 1a. The spheres represent the monomers of size $\sigma$, which we set as the unit of length in the system. All distances are shown with respect to the central monomer (shown in pink). These monomers are allowed to interact with each other using three potentials, a two-body $V_2$, three-body $V_3$ and a four-body potential $V_4$. The behavior of the three potentials is shown in Figure 1b and the potentials are expressed as follows,

- $V_2$: **Two body attractive potential**
  For any two monomers at a distance of $r_2$, an attractive Lennard–Jones potential is provided which is modified by an exponential term as shown in Equation (1).

$$V_2 = \epsilon[(\frac{\sigma}{r_2})^{12} - (\frac{\sigma}{r_2})^6 + \epsilon_1 e^{-ar_2/\sigma}]; \ \forall r_2 < r_c. \tag{1}$$

  where $\epsilon = 110 k_B T$ and the cutoff distance is $r_c = 2.5\sigma$. The exponential term in the above potential creates a maximum at $r_2 = 1.75\sigma$ which acts as a potential barrier for joining or breaking of monomers from chains. The value of $\epsilon_1$ and $a$ are kept fixed as $\epsilon_1 = 1.34\epsilon$ and $a = 1.72$. This potential behavior is shown in Figure 1b where the Y-axis is $V_2 + V_3$. When $\theta = 0$, $V_3 = 0$ (see below). Therefore, the graph shown by the symbols (blue-triangle) having legends $\sin^2 \theta = 0$ represents the behavior of $V_2$. In the graph, $r_2$ is kept fixed at $r_2 = \sigma$ (except for the inset figure).
- $V_3$: **Three body potential to add semi-flexibility to chains**

For any monomer that is part of a chain, there are two bonded neighbours at a distance of $r_3$ and $r_4$ which subtends an angle $\theta$ at the central monomer (as shown in Figure 1). The triplet thus formed is then subjected to the following three-body potential,

$$V_3 = \epsilon_3(1 - \frac{r_2}{\sigma_3})^2(1 - \frac{r_3}{\sigma_3})^2 \sin^2(\theta); \; \forall r_2, r_3 < \sigma_3. \tag{2}$$

where the value of $\epsilon_3 = 6075 k_B T$ and the cutoff distance $\sigma_3$ is kept fixed at $1.5\sigma$. The leading terms inside the two brackets ensure that the potential and force goes smoothly to zero at the cutoff of $\sigma_3$.

- $V_4$: **Four body repulsive potential between chains**
  For any monomer with two bonded neighbours at distances $r_2$ and $r_3$, any other monomer at a distance $r_4$ approaching the first monomer to form a branch (see Figure 1a) will be repelled with the following potential,

$$V_4 = \epsilon_4(1 - \frac{r_2}{\sigma_3})^2(1 - \frac{r_3}{\sigma_3})^2 \times V_{LJ}(\sigma_4, r_4) \tag{3}$$

The cutoff distance for this potential $\sigma_4$ is chosen such that $\sigma_3 < \sigma_4 < r_c$ and is fixed at $\sigma_4 = 1.75\sigma$. The leading terms in the brackets are necessary to make the force and potential smoothly approaching zero at the cutoff distance. Since, those terms in the brackets approaches zero as $r_2$ or $r_3$ approaches $\sigma_3$, therefore the value of $\epsilon_4$ is decided to give a very high value $\epsilon_4 = 2.53 \times 10^5 \; k_B T$ to ensure enough repulsion between the chains. The behavior of $V_4$ is shown in the inset of Figure 1b. It should be noted that we refer to micellar chains as dispersed if the distance between chains is $> 1.75\sigma$. When the distance between chains of monomers is $< 1.75\sigma$, then we refer to them as clusters of chains.

Using Monte Carlo technique, the system is allowed to equilibrate from a randomly initialized state. After around $5 \times 10^5 - 6 \times 10^5$ iterations, the system evolves to form Wormlike chains of monomers having an exponential distribution of chain length [48]. With an increase in micellar density, an isotropic-to-nematic transition is observed, as has been reported in detail [48].

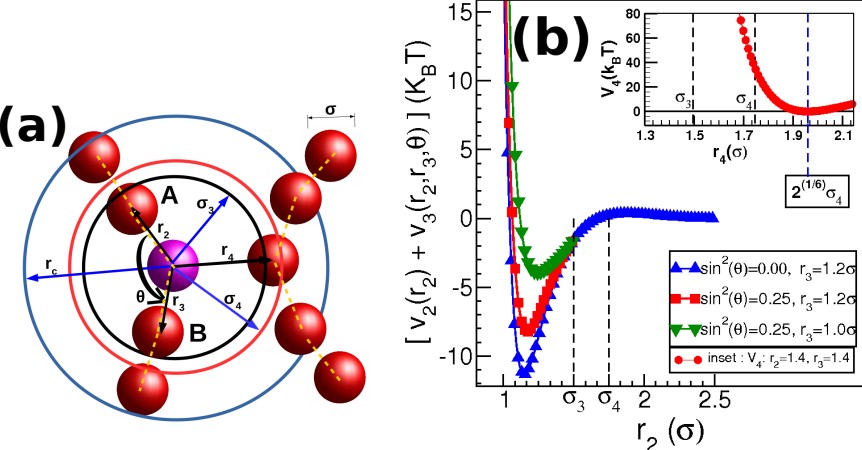

**Figure 1.** The figure shows the modeling of the Wormlike micelles. The spheres denote the micellar monomers having diameter $\sigma$. All the distances are measured with respect to the central monomer shown in pink. These monomers are acted upon by two body potential $V_2$ having a cutoff range of $r_c$. A three-body potential $V_3$ is acting on a triplet with a central monomer (pink) bonded with two other monomers at distances of $r_2$ and $r_3$, forming an angle $\theta$ at the central monomer and having a cutoff range of $\sigma_3$. In addition to these potentials, there exists a four-body potential $V_4$ which is a shifted Lennard–Jones potential introduced to prevent branching and having a cutoff distance $2^{1/6}\sigma_4$.

## 2.2. Modeling Nanoparticles

To investigate the phase behavior of Wormlike micelle-nanoparticle system, model nanoparticles are added in the model Wormlike micellar system described in the above section. Nanoparticles are modeled by Lennard–Jones attractive particles of size $\sigma_n$ and having a cutoff distance $r_{cn}$ with the interacting potential $V_{2n}$ given by,

$$V_{2n} = \epsilon_n [(\frac{\sigma_n}{r_n})^{12} - (\frac{\sigma_n}{r_n})^6], \forall r_n <= r_{cn} \tag{4}$$

The cutoff distance $r_{cn}$ is set at $r_{cn} = 2\sigma_n$. These nanoparticles interact with monomers via a repelling potential $V_{4n}$ which is a shifted Lennard–Jones potential given by,

$$V_{4n} = \epsilon_{4n} [(\frac{\sigma_{4n}}{r_{mn}})^{12} - (\frac{\sigma_{4n}}{r_{mn}})^6], \forall r_{mn} <= 2^{1/6}\sigma_{4n} \tag{5}$$

where $r_{mn}$ indicates the distance between monomers and nanoparticles with the parameter $\sigma_{4n}$ indicating the centre-to-centre distance between the particles. The value of $\sigma_{4n}$ is used as a parameter. The value of the strength of the repulsive interaction is fixed at $\epsilon_{4n} = 30k_BT$.

In summary, in this model (refer Figure 1a), there are coarse-grained particles (spheres) to form micellar chains of the size $\sigma$ that interacts with each other via a Lennard–Jones potential having a cutoff distance $2.5\sigma$. The micellar chains are semi-flexible (potential $V_3$) and repel each other with the potential $V_4$ and the minimum approaching distance $\sigma_4 = 1.75\sigma$. The system also contains nanoparticles of size $\sigma_n$ which are interacting by a Lennard–Jones attractive potential among themselves (potential $V_{2n}$) with a cutoff distance $2\sigma_n$. These monomers and nanoparticles are repelled by a repulsive potential $V_{4n}$ with the minimum approaching distance $\sigma_{4n}$ and cutoff distance $2^{1/6}\sigma_{4n}$. The value of $\sigma_{4n}$ is used as a parameter.

Using this model, the morphological transformations of the system with a change in $\sigma_{4n}$ and micellar density has been established [47] (keeping the nanoparticle size fixed at $\sigma_n = 1.5\sigma$). Now, in this paper, the system is investigated to explore the effect of the strength of the attractive interaction between nanoparticles $\epsilon_n$ on the system behavior. Therefore, we keep the nanoparticle size fixed at $\sigma_n = 1.5\sigma$, while using $\epsilon_n$ as a variable along with $\sigma_{4n}$ and the number density of monomers $\rho_m$. We generate runs in sets where each set consists of runs with a fixed $\epsilon_n$ but $\sigma_{4n}$ and $\rho_m$ as parameters.

## 2.3. Method

The model is first applied with the Metropolis Monte Carlo (MC) method. However, the method seemed to be insufficient to equilibrate the system with high density. Therefore, the system is first evolved with Metropolis Monte Carlo method with 200–300 nanoparticles within a given number density of monomers $\rho_m$ for $10^5$ iterations. This gives the monomers enough time to develop into chain-like structures in the presence of seeding of nanoparticles. Then, a semi-grand canonical Monte Carlo (GCMC) scheme is applied. According to this scheme, for every 50 Monte Carlo steps, 300 attempts are made to add and remove a nanoparticle randomly. Each successful attempt is penalized with an energy gain or loss of $\pm \mu_n$, where, $\mu_n$ is the chemical potential of the system fixed at $\mu_n = -8 k_BT$. All runs were tested with ten independent runs which show convergence to morphologically similar states and their thermodynamic properties converging to the same values. It is shown that, for the model system, runs that started with an unmixed state (both nanoparticles and monomers separated) also tend to form a mixed state for the value of $\mu_n$ chosen. Thus, the possibility of a fully phase separated state as a thermodynamically preferred state is negated [47].

For all the runs in this paper, we first apply the Metropolis Monte Carlo method to allow the growth of equilibrium polymeric chains in the presence of seed of nanoparticles. Then, GCMC scheme is switched on for the rest of the run. For each set of parameters, the system is evolved for $\approx 20 \times 10^5 - 40 \times 10^5$ iterations and the thermodynamic properties are averaged for the last

$10 \times 10^5 - 20 \times 10^5$ iterations over ten independent runs. The error bars in the plots shown in this paper are smaller than the symbols and hence not visible here.

## 3. Results

Previous studies [47,48] using the same model (presented in the previous section) have reported the morphological transitions of nanoparticles with the increase in $\sigma_{4n}$. Those results were substantiated by showing the convergence of all ten independent runs to same morphological structures. It is emphasized here that all these studies are comprised of systems that are initialized with a mixed state of nanoparticles and micelles. It was shown that the nanoparticle clusters formed in the system vary in their shape anisotropy with a change in matrix polymer density. This paper takes this investigation further by varying the strength of interaction between nanoparticles $\epsilon_n$ along with $\sigma_{4n}$ and monomer No. density $\rho_m$. All quantities calculated are averaged over ten independent runs.

To investigate the effect of the strength of interaction between nanoparticles $\epsilon_n$, a set of runs varying in the value of $\rho_m$ and $\sigma_{4n}$ is generated for each value of $\epsilon_n$. For a given value of $\epsilon_n$, the system morphological behavior is observed and the structural changes are identified. Then, these structural changes are compared over different values of $\epsilon_n$ and the change in the value of EVP at which the morphological change occurs are observed. Different values of $\epsilon_n = 2k_BT, 5k_BT, 8k_BT$ and $11k_BT$ are considered. Apart from these values, one more case where there exists no attractive interaction between nanoparticles is also considered. In this case, the nanoparticles are provided with WAC (Week–Anderson–Chandler) potential (similar to the potential expressed in Equation (5). We represent this case by $\epsilon_n = 0$ only for convenience. For each value of $\epsilon_n$, a set of runs with four values of number density of monomers $\rho_m = 0.037\sigma^{-3}, 0.074\sigma^{-3}, 0.093\sigma^{-3}$ and $0.126\sigma^{-3}$, along with varying parameter $\sigma_{4n}$ for each density, are produced. The system is evolved using MC steps for first $10^5$ iterations and then subjected to GCMC scheme for the rest of the iterations. The system is monitored to ensure that the runs are long enough to produce structures and thermodynamic quantities that are stable over a long run. After around $2 \times 10^5 - 3 \times 10^5$ iterations, the systems are observed to maintain their morphological states. The behavior of the average energy of the particles for $\rho_m = 0.037\sigma^{-3}$ is shown in Figure 2 for two different values of $\epsilon_n$: (a) 0; and (b) $2k_BT$. For each value of $\epsilon_n$, the figure shows graphs for four different values of $\sigma_{4n} = 1.5\sigma, 1.75\sigma, 2\sigma$ and $2.25\sigma$. In both figures, all graphs show a jump in their energy values at $10^5$ Monte Carlo Steps (MCSs). These jumps mark the starting of the GCMC scheme where nanoparticles start getting introduced into the system. After around $2 \times 10^5$ MCSs, the system morphology is observed to remain the same. With the increase in the value of $\epsilon_n$, the system becomes very dense. Therefore, the systems seem to be stuck in some kinetically arrested states for $\epsilon_n > 0$. This can be seen in Figure 2, which shows the evolution of the number of nanoparticles in the simulation box with MCSs for: (a) $\epsilon_n = 0$; and (b) $\epsilon_n = 2k_BT$. After some MCSs, the number of nanoparticle and energy graphs show a very slow increase in its value for $\epsilon_n = 2k_BT$ in Figures 3b and 2b, respectively. Only for $\epsilon_n = 0$, the system shows a stable value of energy and the number of nanoparticles as shown in Figures 2a and 3a. Therefore, the systems with higher values of $\epsilon_n$ seem to be in a kinetically arrested state. However, for each of the values of $\epsilon_n$, the ten independent runs converge to the same value of energy and morphological structure.

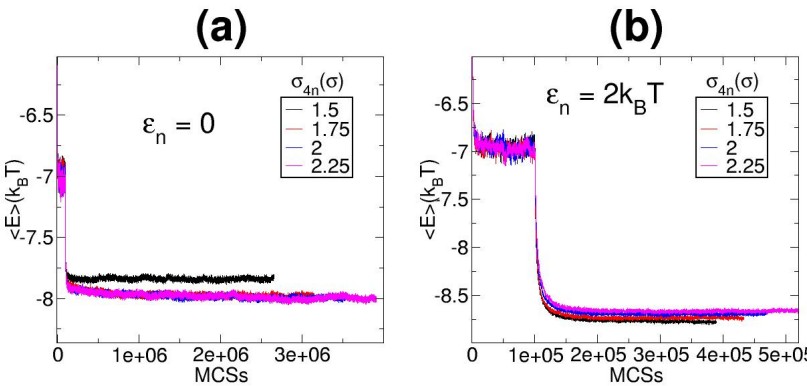

**Figure 2.** The figure shows the evolution of the average energy of the system with Monte Carlo steps for the values: (**a**) $\epsilon_n = 0$; and (**b**) $\epsilon_n = 2\,k_BT$. Each figure shows graphs for four different values of $\sigma_{4n} = 1.5\sigma$, $1.75\sigma$, $2\sigma$ and $2.25\sigma$. All graphs show a jump in their values at $10^5$ Monte Carlo steps indicating the start of GCMC scheme.

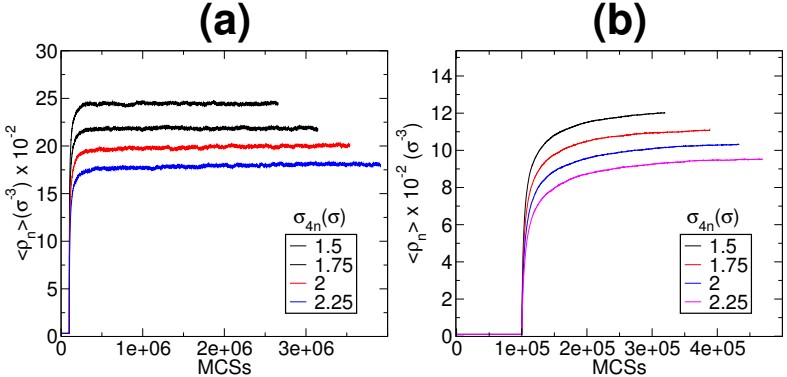

**Figure 3.** The figure shows the evolution of the volume fraction of nanoparticles with Monte Carlo steps for values: (**a**) $\epsilon_n = 0$; and (**b**) $\epsilon_n = 2\,k_BT$. Each figure shows the graphs of nanoparticle volume fraction for four different values of $\sigma_{4n} = 1.5\sigma$, $1.75\sigma$, $2\sigma$ and $2.25\sigma$. All graphs show a jump at $10^5$ Monte Carlo steps when the GCMC scheme is switched on.

The observed behavior is found to be similar for $\rho_m = 0.074\sigma^{-3}$, $0.093\sigma^{-3}$ and $0.126\sigma^{-3}$. Therefore, only the snapshots for $\rho_m = 0.093\sigma^{-3}$ are used here to illustrate the behavior for all these densities. Few snapshots for illustration for other densities are also shown at the end. Figures 4 and 5 show the snapshots for values of $\epsilon_n = 2k_BT$ and $11k_BT$, respectively. Each figure shows snapshots for different values of $\sigma_{4n}$ increasing from(a–d) (or (e–h)). The upper row shows both the micelles (red particles) and nanoparticle (blue), while the lower row shows only nanoparticles. For the size of nanoparticle $\sigma_n = 1.5\sigma$ considered here, the minimum value of $\sigma_{4n}$ is $1.25\sigma$ [47]. For this value of $\sigma_{4n} = 1.25\sigma$, the micellar chains and nanoparticles form a uniformly mixed state (Figures 4a and 5a). No two micellar chains are found without nanoparticles in between (i.e., no clustering of chains). An increase in the value of $\sigma_{4n}$ from $1.25\sigma$ leads to the formation of clusters of micellar chains that forms a network-like structure as shown in Figures 4b and 5b. Therefore, nanoparticles in (Figures 4f and 5f) are also forming network-like structures. With further increase in the value of $\sigma_{4n}$, the nanoparticle network start breaking (Figures 4g and 5g) and finally the network of nanoparticles breaks into individual clusters (Figures 4h and 5h).

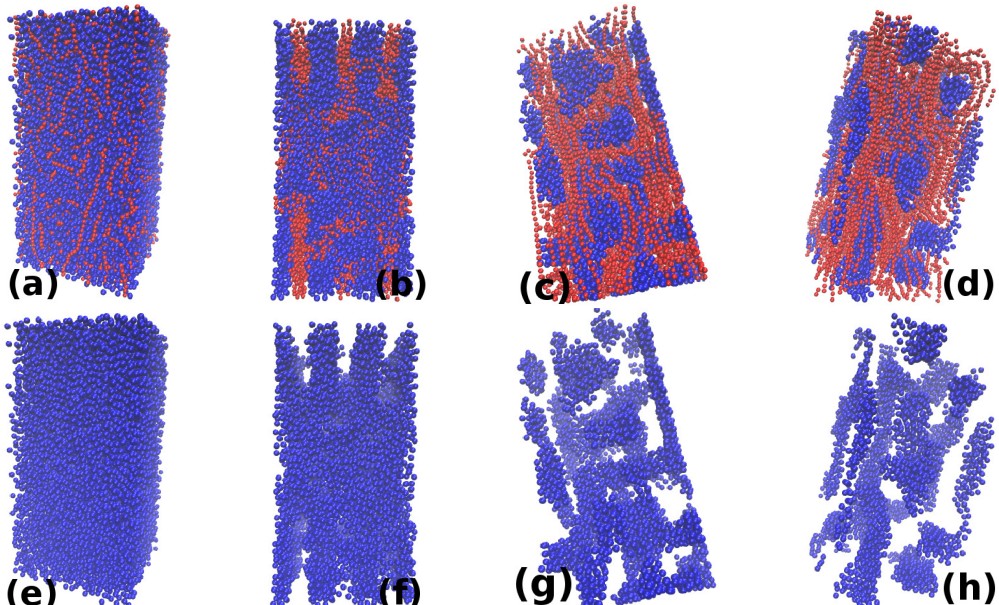

**Figure 4.** The figure shows snapshots for four different values of $\sigma_{4n}$: (**a**) $1.25\sigma$; (**b**) $1.75\sigma$; (**c**) $2.5\sigma$; and (**d**) $2.75\sigma$ for $\epsilon_n = 2k_BT$. The upper row shows both the nanoparticles and monomers, while the lower row shows only nanoparticles. For $\sigma_{4n} = 1.25\sigma$, the micellar chains form a dispersed state. For $\sigma_{4n} > 1.25\sigma$, the nanoparticles and micellar chains form interpenetrating network-like structures which show a morphological transition for $\sigma_{4n} = 2.75\sigma$ forming individual sheets of nanoparticles.

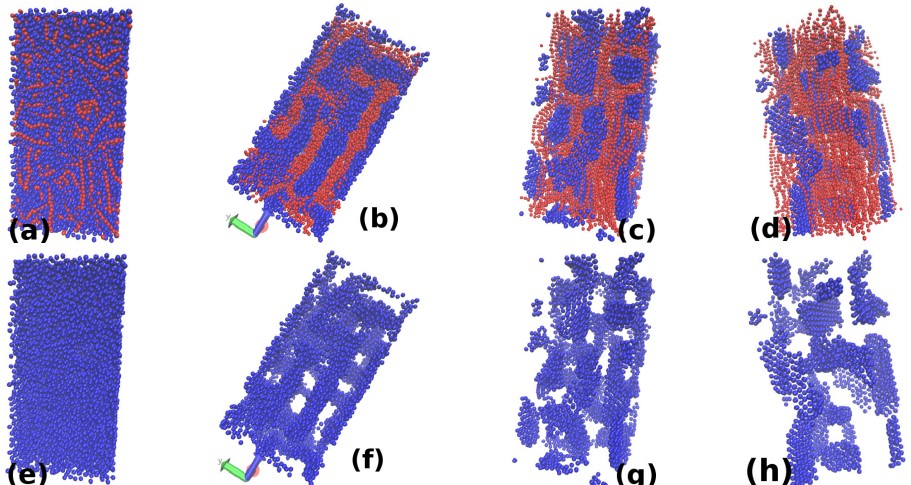

**Figure 5.** The figure shows snapshots for four different values: $\sigma_{4n}$ is $1.25\sigma, 2\sigma, 2.5\sigma$ and $3\sigma$ in (**a**–**d**), $\sigma_{4n}$ is $1.25\sigma, 2\sigma, 2.5\sigma$ and $3\sigma$ in (**e**–**h**), respectively, for $\epsilon_n = 11k_BT$. The upper row shows both the nanoparticles and monomers, while the lower row shows only nanoparticles. The polymeric chains at $\sigma_{4n} = 1.25\sigma$ form a dispersed state. For $\sigma_{4n} > 1.25\sigma$, the nanoparticles and micellar chains form interpenetrating networks which show a morphological change for $\sigma_{4n} = 2.75\sigma$ forming individual sheets of nanoparticles.

Similar morphological changes are observed for other values of $\rho_m$, but with a difference in the anisotropy of nanoparticle clusters. For $\rho_m = 0.093\sigma^{-3}$, the nanoparticles form sheet-like structures, while, for $\rho_m = 0.126\sigma^{-3}$, rod-like nanoparticle clusters are observed. However, irrespective of the value of $\epsilon_n$, similar morphological transitions are observed for all $\epsilon_n$. This shows that changes in the morphology of the structures are due to change in $\sigma_{4n}$ only. Moreover, the sheet-like morphology of the structures formed in Figures 4h and 5h verifies the previous result that the micellar density governs the

morphology of nanoparticle structures. For $\epsilon_n = 0$, however, without attractive interaction between nanoparticles, a nanostructure cannot be formed; it only shows the arrangement of nanoparticles mediated by the micellar matrix. Thus, the value of $\epsilon_n$ does not seem to be affecting the morphological behavior of the system. However, the change in $\epsilon_n$ affects two things: the value of $\sigma_{4n}$ at which the morphological change in nanoparticle structure occurs and the packing of nanoparticles.

　　Examining the Figures 4 and 5, we see that the value of $\sigma_{4n}$ at which the nanoparticles undergo a transition from network-like morphology to individual clusters of nanoparticles, gets shifted to a higher value of $\sigma_{4n}$ with an increase in $\epsilon_n$. For $\epsilon_n = 0$, the breaking of network into individual clusters occurs at $\sigma_{4n} = 2.5\sigma$ (see the Supplementary Materials), but this happens at $\sigma_{4n} = 2.75\sigma$ and $3\sigma$ for the value of $\epsilon_n = 2k_BT$ (or $5k_BT$) and $11k_BT$, respectively (Figures 4h and 5h). This is because nanoparticle density increases with an increase in $\epsilon_n$, but decreases with an increase in the value of $\sigma_{4n}$. Therefore, to reach the low nanoparticle density required to produce individual clusters, systems with a higher value of $\epsilon_n$ need to get higher values of $\sigma_{4n}$. Although the value of EVP for change in nanoparticle structure gets shifted, the value of EVP for the change in micellar chains structure from dispersed state to the formation of clusters (for $\sigma_{4n} = 1.25\sigma$ to $1.5\sigma$) does not change with the change in $\epsilon_n$.

　　An increase in the value of $\sigma_{4n}$ demands an increase in the excluded volume of the system. Therefore, with increase in $\sigma_{4n}$ from $1.25\sigma$ to $1.5\sigma$, the system reorganizes itself to lower its excluded volume due to both $V_4$ and $V_{4n}$. This reorganization evokes a competition between the excluded volume due to $V_4$ and the excluded volume due to $V_{4n}$. When the value of $\sigma_{4n}$ increases from $1.25\sigma$ to $1.5\sigma$, then the micellar chains reorganizes to form clusters to decrease $V_{4n}$, but increasing $V_4$. This is because the number (or energy) of nanoparticles is higher than monomers (as shown in the following sections). When the value of $\sigma_{4n}$ increases to a high value such that decreasing the distance between chains is more costly in terms of energy, then the nanoparticle density is decreased and the nanoparticle network starts breaking. When the nanoparticle network breaks to an extent that micellar chains get enough volume to be away from each other's repulsive interaction ($V_4$) range ($2^{1/6}\sigma_4$), then the total excluded volume of the system decreases. This change in the distance between micellar chains can be confirmed by plotting the pair correlation function. This is shown in Figure 6a.

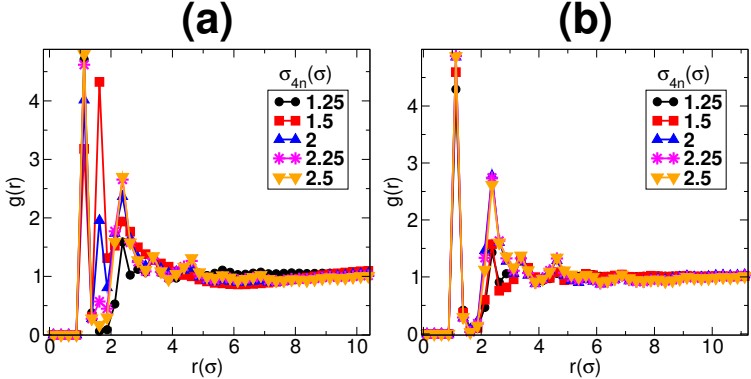

**Figure 6.** The figure shows the pair correlation function of monomers for values of $\epsilon_n$: (**a**) $2k_BT$; and (**b**) 0. Each figure shows five different graphs for values of $\sigma_{4n} = 1.25\sigma$, $1.5\sigma$, $2\sigma$, $2.25\sigma$ and $2.5\sigma$. A peaks around $1.75\sigma$ indicates clustering of micellar chains. For $\epsilon_n = 8k_BT$, there appear peaks around $1.75\sigma$, which decreases with increase in $\sigma_{4n}$. However, there are no peaks around $1.75\sigma$ for $\epsilon_n = 0$.

　　The Figure 6a shows plots of monomer pair correlation function for $\epsilon_n = 2k_BT$. The behavior of the pair correlation function for monomers is similar for all $\epsilon_n > 0$. Figure 6a shows graphs for five different values of $\sigma_{4n} = 1.25\sigma$, $1.5\sigma$, $2\sigma$, $2.25\sigma$ and $2.5\sigma$. For monomer pair correlation function, a first peak is expected at $\approx \sigma$ and its multiples indicating the monomers which are part of a chain. Peaks are also expected to occur at $1.75\sigma$ and its multiples if the chains are within the range of repulsive potential $V_4$. In the figure, the pair correlation function for $\sigma_{4n} = 1.25\sigma$ does not show any peak around

$1.75\sigma$. This shows that the chains are dispersed in between nanoparticles, as shown in the snapshots in Figures 4a and 5a. With the increase in $\sigma_{4n}$ from $1.25\sigma$, there appear peaks around $1.75\sigma$ and its multiples. This indicates the formation of clusters of micellar chains that join to form a network, as shown in Figures 4b and 5b. However, this peak decreases in its height with the increase in $\sigma_{4n}$. This decrease is due to the decrease in the density of nanoparticles or breaking of the nanoparticle network that decreases the excluded volume between micellar chains. For $\sigma_{4n} = 2.75\sigma$, the network breaks to an extent that micellar chains get enough volume to be out of the range of the repulsive interaction ($V_4$) from each other.

If nanoparticle energy is lower (-ve)than micellar chains, then an increase in $\sigma_{4n}$ will lead to more clustering of micellar chains such that the distance between micellar chains decreases while the distance between micellar chains and nanoparticles increases (micellar chains are "pushed" by nanoparticles). However, if nanoparticle energy is higher, then nanoparticle density decreases without any decrease in the distance between micellar chains. This competition between the energies of nanoparticles and micellar chains can be clearly observed by examining the pair correlation functions for different values of $\epsilon_n$. Figure 6b represents the monomer pair correlation function for $\epsilon_n = 0$. Comparing Figure 6a,b, we can clearly see that there are no peaks around $1.75\sigma$ for $\epsilon_n = 0$. This clearly shows that the energy of nanoparticles in case of $\epsilon_n = 0$ is not competitive with monomers and, hence, micellar chains do not form clusters. Therefore, for $\epsilon_n = 0$, with the increase in $\sigma_{4n}$, the number of nanoparticles decreases (or the nanoparticle network breaks) without decreasing the distance between micellar chains.

Thus, we see that, with the increase in $\sigma_{4n}$, the total excluded volume of the system increases, as a result of which the system reorganizes itself. Therefore, it is realized that the behavior of the system can be explained well if we take into account the excluded volume in the system. To take into account the excluded volume, the volume of the matrix polymeric chains are described along with the excluded volume which we call as the effective volume of micelles (or monomers). The effective volume of monomers $V_m{}^{eff}$ is defined as the total excluded volume due to repulsive interactions between chains of monomers $V_4$ and in between the monomers and nanoparticles $V_{4n}$ in addition to the volume of monomers. The scheme to calculate the effective volume of micelles (or EPs) is shown in Figure 7. The figure shows that any two micellar chains at a distance $r < \sigma_4$ ($\sigma_4$, the cutoff distance for $V_4$) are considered as cylinders of diameter $\sigma_4$, while any monomer at a distance $r < \sigma_{4n}$ ($\sigma_{4n}$, cutoff distance for $V_{4n}$) is considered as a sphere of radius $\sigma_{4n} - \sigma_n/2$.

To calculate the effective volume of matrix polymers, a suitable algorithm is used to first sort out monomers which are part of a single chain. Then, all chains involved in the repulsive interaction $V_4$ with other chains or repelling a nanoparticle with $V_{4n}$ are found. Then, using the scheme explained in Figure 7, the effective volume of chains is calculated. This effective volume not only depends on the value of $\sigma_{4n}$ but also on the arrangement of the constituent particles that determines the number of pairs of particles repelling each other. This, in turn, depends on the density of nanoparticles. Using the scheme shown in Figure 7, the effective monomer volume fraction might be slightly overestimated, but that is insignificant and does not affect the results.

The behavior of the effective volume fraction of monomers and nanoparticles is shown in Figure 8a,b, respectively. Each graphs shows different values of $\epsilon_n$. It should be noted that for a given value of $\sigma_{4n}$, a high value of the effective volume of monomers indicates the presence of a large number of pairs of particles having repulsive interactions between monomer chains or between monomers and nanoparticles. In Figure 8a, the effective volume shows an increase in its value with an increase in $\sigma_{4n}$ from $1.25\sigma$, representing the change from a dispersed state of chains to its clusters. Then, it shows a decrease in its value for further increase in $\sigma_{4n}$ for all the values of $\epsilon_n$ except $\epsilon_n = 11k_BT$. A decrease in the value of $V_m{}^{eff}$ shows the breaking of nanoparticle network to the extent that micellar chains get enough volume to be away from each other's repulsive interaction range. We see that only for $\epsilon_n = 11k_BT$, the value of effective volume keeps on increasing with an increase in $\sigma_{4n}$ and then shows a decrease at a higher value of $\sigma_{4n} = 3\sigma$. The changes in $V_m{}^{eff}$ are insignificant for $\epsilon_n = 0$ and it shows a nearly constant low value of $V_m{}^{eff}$. This confirms that, in the case of $\epsilon_n = 0$, an increase in $\sigma_{4n}$

leads to a decrease in the number of nanoparticles without decreasing the distance between micellar chains. In the graphs for nanoparticle volume fraction in Figure 8b, one can see that the nanoparticle volume fraction not only decreases with increase in $\sigma_{4n}$ but also decreases slightly for a decrease in $\epsilon_n$. The plots show similar values for all $\epsilon_n > 0$ but distinctly lower values for the case of $\epsilon_n = 0$. This indicates that, in the case of $\epsilon_n = 0$, the nanoparticle volume fraction is much lower than that of $\epsilon_n > 0$. Hence, the transition from network to individual clusters of nanoparticles can be reached at a much lower value in case of $\epsilon_n = 0$. As explained above, the increase in the excluded volume is because of the competition between the energies of the clusters of nanoparticles and monomer chains. Therefore, the behavior of the $V_m^{eff}$ can be confirmed by plotting the average energies of the nanoparticles and monomers.

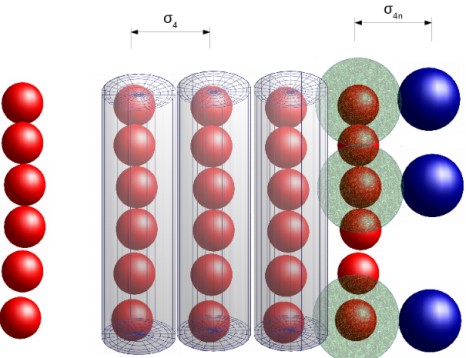

**Figure 7.** The figure explains the calculation of the effective volume of micelles. If any two micellar chains are at a distance $r <= \sigma_4$ from each other, they are considered as cylinders of radius $\sigma_4/2$ shown as a shaded region (red). When a nanoparticle is at a distance $r <= \sigma_{4n}$ from a monomer, then the monomer is assumed as a sphere of radius $\sigma_{4n} - \sigma_n/2$.

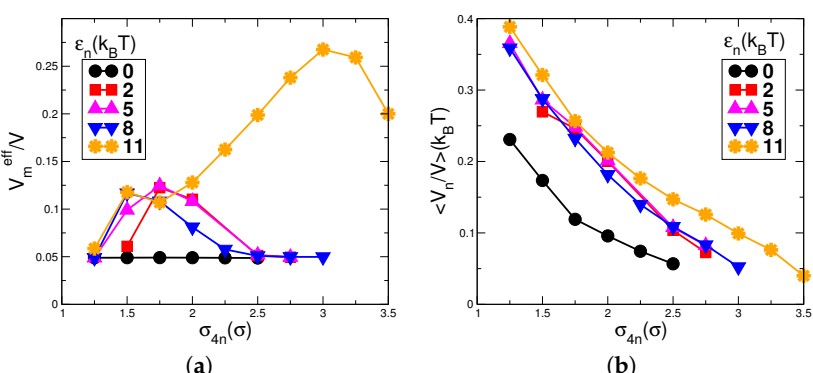

**Figure 8.** (**a**) The effective volume fraction of monomers; and (**b**) volume fraction of nanoparticles. The effective volume of micelles shows an increase in its value with an increase in $\sigma_{4n}$ marking the change from a dispersed state to the formation of clusters of chains. With further increase in $\sigma_{4n}$, it decreases for all values of $\epsilon_n$ except for $\epsilon_n = 11k_BT$ indicating the presence of a high competition between the clusters of nanoparticles and monomer chains. The changes in $\epsilon_n = 0$ are of the very low order of magnitude and appear to be constant in this graph. The nanoparticle volume fraction shows a decreasing behavior with an increase in the value of $\sigma_{4n}$ for all the values of $\epsilon_n$. For $\epsilon_n = 0$, the values of nanoparticle volume fractions are lower than the values for other $\epsilon_n$.

The plots of average energies of monomers and nanoparticles are shown in Figure 9a,b, respectively. Figure 9c shows the plots for both the energies simultaneously. Each figure represents graphs for different values of $\epsilon_n$. Except for $\epsilon_n = 0$, all other values of $\epsilon_n$ indicate the two transformation points in the morphology of the system by showing a non-monotonic behavior in

Figure 9a. With the increase in $\sigma_{4n}$ from $1.25\sigma$ to a higher value, the monomer energy shows an increase in its value while the nanoparticle energy shows a decrease in its value. The increase in energy of monomers is due to the increased repulsive interaction between chains due to clustering of monomer chains. This point corresponds to the transformation from a dispersed state of chains to network-like structures. With further increase in $\sigma_{4n}$ the nanoparticle network breaks (volume fraction of nanoparticle decreases). Due to the breaking of the network, the available volume for monomer chains increase and hence their distance from each other increases, which results in a decrease in their repulsive interaction. Hence, the monomer energy shows a decrease (more -ve) in its value. For a higher value of $\sigma_{4n}$ (depending on $\epsilon_n$), the energy of monomers again show an increase. This increase in energy is due to a decrease in the effective volume of monomers because of the breaking of the nanoparticle network. This leads to a lower chain length of monomers hence increasing their energy [47]. Comparing the energies of monomers and nanoparticles, it can be seen that only the values of energies of monomers and nanoparticles for $\epsilon_n = 11k_BT$ are relatively comparable. For lower values of $\epsilon_n$, nanoparticle energy is higher compared to monomers, as shown in Figure 9c, and the gap between them increases with the decrease in $\epsilon_n$. This confirms the observed behavior of the effective volume of monomers for different values of $\epsilon_n$ shown in Figure 8a.

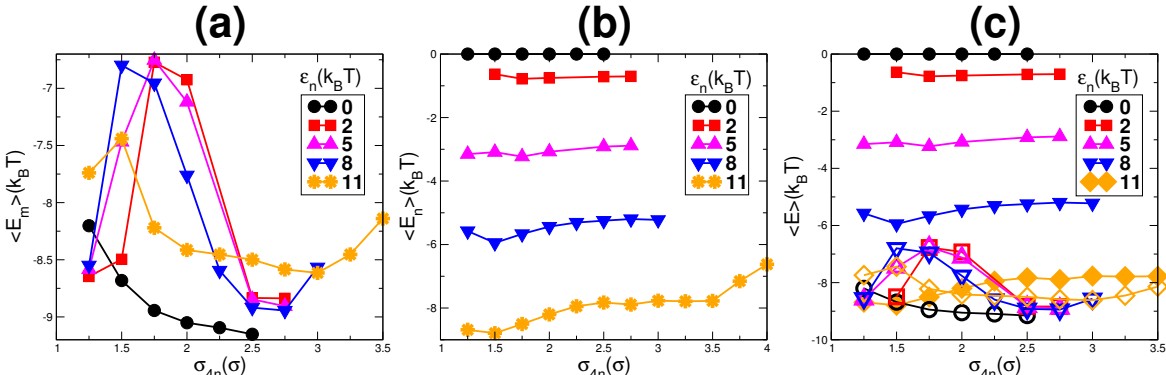

**Figure 9.** The average energy of: (**a**) monomers; (**b**) nanoparticles; and (**c**) both (monomers are empty symbols and nanoparticles are filled symbols). Comparing the two energies, it can be seen that only for $\epsilon_n = 11k_BT$, the energies of monomers and nanoparticles are competitive. For other values of $\epsilon_n$, the gap between the energies is higher, as shown in (**c**).

Apart from the shift in the morphological transformation point (the value of EVP) with the change in $\epsilon_n$, one more change with the change in $\epsilon_n$ can be easily noticed. With the decrease in the value of $\epsilon_n$, the nanoparticle packing decreases. To gain insight into the arrangement of nanoparticles, the pair correlation function g(r) for nanoparticles is plotted in Figure 10. It shows the pair correlation function for nanoparticles with $\sigma_{4n} = 2.5\sigma$ and different values of $\epsilon_n = 0$, $2k_BT$, $5k_BT$ and $11k_BT$, as indicated by the symbols. With the decrease in $\epsilon_n$, the height of the peak decreases. For $\epsilon_n = 0$, the peaks are broader and have a relatively short range of correlation. Hence, the packing of the nanoparticles is lowest in the case of $\epsilon_n = 0$.

As shown above [47], the same kind of behavior is shown by all densities of micelles, $\rho_m = 0.074\sigma^{-3}$, $0.093\sigma^{-3}$ and $0.126\sigma^{-3}$ except for $\rho_m = 0.037\sigma^{-3}$. In the case of $\rho_m = 0.037\sigma^{-3}$, a change in $\sigma_{4n}$ from $1.25\sigma$ to $1.5\sigma$ leads to the clustering of micellar chains which joins to form a network-like structure similar to other densities. However, no change from the network to individual clusters of nanoparticles is observed for this micellar density. For all values of $\sigma_{4n} > 1.25\sigma$, the system shows the formation of a network of nanoparticle clusters and micellar chains with no further structural change observed for any value of $\sigma_{4n}$ considered here. This is true for all $\epsilon_n$ considered here. A comparison of the systems with $\rho_m = 0.037\sigma^{-3}$ for different values of $\epsilon_n$ is shown in Supplementary Materials. The systems for $\rho_m = 0.037\sigma^{-3}$ are also reproduced in a bigger box size of $60 \times 60 \times 60\sigma^3$ to check the simulation artefacts. One of the snapshots for $\sigma_{4n} = 2.75\sigma$ and $\epsilon_n = 0$ is shown in

Figure 11a,b. The snapshot in Figure 11a shows the nanoparticles (blue) and micelles (red) both while, only nanoparticles from the snapshot in Figure 11a are shown in Figure 11b. The snapshots show the network of nanoparticle clusters interpenetrating with the network of micellar chains.

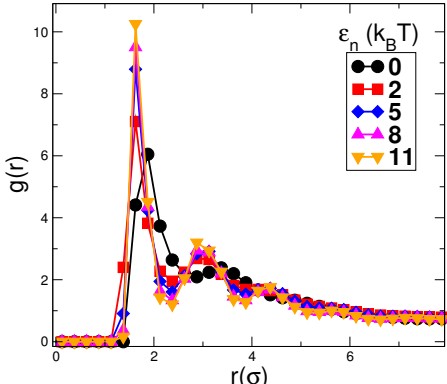

**Figure 10.** The figure shows the pair correlation function for nanoparticles with $\sigma_{4n} = 2.5\sigma$ and for different values of $\epsilon_n$. The height of the peaks decreases with the decrease in $\epsilon_n$. The lowest and broader peak for $\epsilon_n = 0$ shows that the nanoparticle packing, in this case, is lowest.

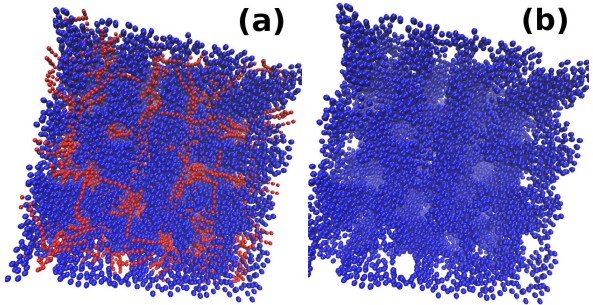

**Figure 11.** Snapshots for $\rho_m = 0.037\sigma^{-3}, \epsilon_n = 0, \sigma_{4n} = 2.75\sigma$ reproduced in a larger box size of $60 \times 60 \times 60\sigma^3$: (**a**) both the nanoparticles (blue) and monomers (red); and (**b**) only nanoparticles. The structure obtained for the larger box size are similar to smaller box size shown in Supplementary Materials.

Throughout the paper, the value of the size of nanoparticles is kept constant at $\sigma_n = 1.5\sigma$. For the range of values of $\sigma_{4n}$ considered, no change from network to individual clusters for nanoparticles is observed for the value of $\rho_m = 0.037\sigma^{-3}$. However, for nanoparticle size $\sigma_n = 3\sigma$, the transition occurs at the value of $\sigma_{4n} = 3.25\sigma$ for the value of $\epsilon_n = 0$. While keeping the other parameters same but increasing $\epsilon_n$ to $11k_BT$, a percolating network of nanoparticles is observed. This is another example of the shift in the value of EVP for the system morphological change with the change in $\epsilon_n$. This is shown in Figure 12. It is interesting to note that the snapshot in Figure 12a shows system spanning sheets of nanoparticles (arranged in alternate layers of nanoparticles and micellar chains). Apart from that, the arrangement or packing of nanoparticles is relatively low in Figure 12a compared to Figure 12b. A similar example is shown in Figure 13. It shows two snapshots for: (a) $\epsilon_n = 0$; and (b) $\epsilon_n = 11k_BT$, keeping the values of $\rho_m = 0.126\sigma^{-3}$ and $\sigma_{4n} = 2.25\sigma$ and $\sigma_n = 1.5\sigma$ the same for both. The system with this micellar No. density $\rho_m = 0.126\sigma^{-3}$ is shown to be produce rod-like morphology of nanoparticle clusters [47]. Here, the snapshots are shown for the value of $\sigma_{4n}$ past the point of transformation from network to individual clusters. Hence, the systems are showing rodlike structures of nanoparticles. However, the rods in case of $\epsilon_n = 0$ (in Figure 13c) can be seen as thinner compared to the rods in Figure 13d for $\epsilon_n = 11k_BT$. Moreover, one can clearly see the difference in the packing of nanoparticles. The nanoparticles are well packed in Figure 13d compared to Figure 13c.

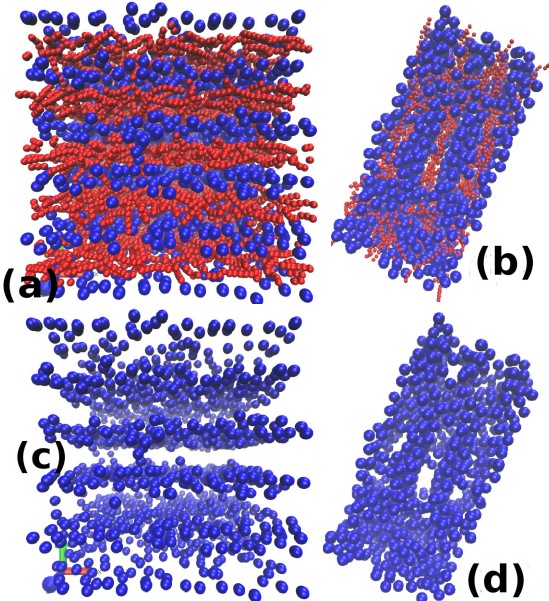

**Figure 12.** Snapshots for $\rho_m = 0.037\sigma^{-3}, \sigma_n = 3\sigma, \sigma_{4n} = 3.25\sigma$ for two different values of $\epsilon_n$: (**a**) 0; and (**b**) $11k_BT$. The upper row shows both the nanoparticles and monomers, while the lower row only shows the nanoparticles. The point of the structural change for nanoparticles gets shifted to lower values of $\sigma_{4n}$ with a decrease in $\epsilon_n$. Therefore, the left snapshots show a system forming system spanning sheet of nanoparticles while the right figure is still in the regime of percolating network-like structure despite having the same values of $\sigma_{4n}$.

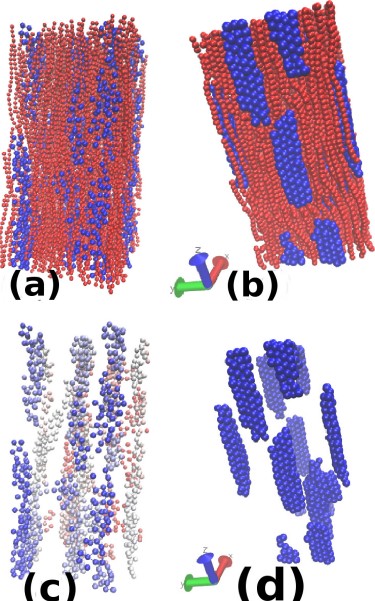

**Figure 13.** Snapshots for the $\rho_m = 0.126\sigma^{-3}$ and $\sigma_{4n} = 2\sigma$: (**a**) $\epsilon_n = 0$; and (**b**) $\epsilon_n = 11\ k_BT$. The upper row only shows both the nanoparticles (blue) and micelles (red) while the lower row only shows nanoparticles. Only (**c**) has a gradient in its colour along one of the shorter axes of the box. Comparison of both the figure shows that a lower value of $\epsilon_n$ leads to low volume fraction and low packing of nanoparticles. Therefore the rods formed in (**c**) do not show a well-packed structure compared to the right figure and are also thinner.

## 4. Conclusion

A detailed investigation of the effect of the strength of interaction between nanoparticles on the structural behavior of the Wormlike micelles-nanoparticles system is carried out. It is shown that with the decrease in the value of $\epsilon_n$, the point (value of EVP) of the transition from network to individual clusters of nanoparticles gets shifted to the lower value of $\sigma_{4n}$. It is also shown that this shift in transition point is due to a decrease in the nanoparticle volume fraction with the decrease in $\epsilon_n$. For the case of $\epsilon_n = 0, \sigma_n = 3\sigma$ and $\sigma_{4n} = 3.25\sigma$, system spanning sheet-like arrangement of nanoparticles is reported. The investigation also shows that a decrease in the value of $\epsilon_n$ leads to a decrease in the packing of nanoparticles.

**Supplementary Materials:** Supplementary Materials can be found at: www.mdpi.com/2410-3896/3/4/31/s1.

**Funding:** This research received no external funding.

**Acknowledgments:** We thank the computational facility provided National Param Super Computing (NPSF) CDAC, India for use of Yuva cluster, the computer cluster in IISER-Pune, funded by DST, India by Project No. SR/NM/NS-42/2009.

**Conflicts of Interest:** The authors declare no conflict of interest.

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
