# Peer review of "Effect of the Strength of Attraction Between Nanoparticles on Wormlike Micelle- Nanoparticle System"

_condensedmatter, doi:10.3390/condmat3040031_

Round 1
Reviewer 1 Report
The manuscript entitled "Effect of the strength of attraction between nanoparticles on Wormlike
micelle-nanoparticle system" has been reviewed. The comments are as follows:
1. Initial capital letters, like Equilibrium, Wormlike should be lower-case letters.
2. Abbreviation, such as EVP, should give the full name where it first appears.
Reviewer 2 Report
Please see my comments in the attached file.

Author Response
Dear sir,
Thank you very much for going through my paper in detail and providing a detailed and an in-depth review. I request you to refer the attached file for the response. Thank you very much for the time invested.
Thank You,
Mubeena.

Reviewer 3 Report
The manuscript by Mubeena reported the role of attraction strength in wormlike micelle systems. The shape of nanoparticles could be controlled by polymer density and author investigated the particle interactions in the micelle formation. Overall the manuscript is meaningful and significant in understanding micelles with various shapes. According to its current shape, I suggest a minor revision. Authors applied MC method in modeling and set the density for 10^5 iterations. The claim is not wrong, however, in developing micelle process, there are two types of self-assembly. Author's claim is based on the model that particles formation at the initial stage and the interaction approaches the wormshape. But this process could be another way around. In the discussion part, it is helpful for authors to specify in discussion part.
Author Response
Dear sir,
Thank you very much for appreciating the results. I request you to go through the attached file to refer to the response. Thank you for your precious time.
Thank You,
Mubeena.

Reviewer 4 Report
This article described the factors that affect the spherical micelles transformation into worm-like micelles. The authors have demonstrated several models in the transformation. However, there is still some questions or suggestions for this articles:
In this study, the inner interactions between micelles including attractive potentials and repelling potentials were mentioned. And I wondered if the driving/repelling force from environment such as the solvent would affect the results.
In the model and method section, the authors have mentioned the previous reports. I suggested that the previous work would also involve in the introduction or discussion.
Author Response
Dear sir,
Thank you for your review on my paper. I request you to refer the attached pdf file to get the response. Thank you for your precious time invested.
Thank You,
Mubeena.

Round 2
Reviewer 2 Report
I've found that the author modified their manuscript extensively to address the points in my comments last time. Therefore, I believe the current version shows the potential to appear in Condensed Matter.